# Digital multiple health behaviour change intervention targeting online help seekers: protocol for the COACH randomised factorial trial

Katarina Åsberg,[1] Jenny Blomqvist,[1] Oskar Lundgren,[1] Hanna Henriksson,[1] Pontus Henriksson,[1] Preben Bendtsen,[1,2] Marie Löf,[1,3] Marcus Bendtsen [1]

**To cite:** Åsberg K, Blomqvist J, Lundgren O, *et al*. Digital multiple health behaviour change intervention targeting online help seekers: protocol for the COACH randomised factorial trial. *BMJ Open* 2022;**12**:e061024. doi:10.1136/bmjopen-2022-061024

[1]Department of Health, Medicine and Caring Sciences, Linköping University, Linköping, Sweden
[2]Department of Medical Specialist, Motala Hospital, Motala, Sweden
[3]Department of Biosciences and Nutrition, Karolinska Institutet, Stockholm, Sweden

**Correspondence to**
Dr Marcus Bendtsen;
marcus.bendtsen@liu.se

## ABSTRACT

**Introduction** Unhealthy lifestyle behaviours continue to be highly prevalent, including alcohol consumption, unhealthy diets, insufficient physical activity and smoking. There is a lack of effective interventions which have a large enough reach into the community to improve public health. Additionally, the common co-occurrence of multiple unhealthy behaviours demands investigation of efforts which address more than single behaviours.

**Methods and analysis** The effects of six components of a novel digital multiple health behaviour change intervention on alcohol consumption, diet, physical activity and smoking (coprimary outcomes) will be estimated in a factorial randomised trial. The components are designed to facilitate behaviour change, for example, through goal setting or increasing motivation, and are either present or absent depending on allocation (ie, six factors with two levels each). The study population will be those seeking help online, recruited through search engines, social media and lifestyle-related websites. Included will be those who are at least 18 years of age and have at least one unhealthy behaviour. An adaptive design will be used to periodically make decisions to continue or stop recruitment, with simulations suggesting a final sample size between 1500 and 2500 participants. Multilevel regression models will be used to analyse behavioural outcomes collected at 2 months and 4 months postrandomisation.

**Ethics and dissemination** Approved by the Swedish Ethical Review Authority on 2021-08-11 (Dnr 2021-02855). Since participation is likely motivated by gaining access to novel support, the main concern is demotivation and opportunity cost if the intervention is found to only exert small effects. Recruitment began on 19 October 2021, with an anticipated recruitment period of 12 months.

**Trial registration number** ISRCTN16420548.

## STRENGTHS AND LIMITATIONS OF THIS STUDY

⇒ Pragmatic recruitment of individuals seeking help online to a factorial trial allow for dismantling of the effectiveness of the components which make up a digital multiple health behaviour change intervention.
⇒ An adaptive trial design reduces the risk of under-recruitment and over-recruitment of participants.
⇒ Despite double blind procedures, research participation effects may affect self-reported outcomes and introduce bias.
⇒ Single face-valid items used to measure mediators reduce participant burden but may limit the interpretation of findings.

respiratory disease, cancer and diabetes.[1 2] The WHO has determined that reducing the prevalence of behavioural risk factors should be a priority in many societies to reduce the incidence of NCDs and disability-adjusted life-years.[3] It is therefore important that effective and scalable means of helping individuals to improve their health behaviours are established.

The Public Health Agency of Sweden's national public health survey from 2020[4] (n=16 947) reports data on lifestyle behaviours of Swedish citizens aged 16–84. According to the survey, 16% of respondents report hazardous or harmful alcohol consumption, 35% report being insufficiently physically active, 12% report smoking occasionally or daily and 93% report eating less fruit and vegetables than recommended. Additionally, 52% of individuals report being obese or overweight. Unfortunately, with the exception of smoking, the prevalence rates of these behaviours have not decreased markedly over the past 10 years, with some increasing, witnessing of a lost decade for prevention efforts.

## INTRODUCTION

Behavioural risk factors, such as harmful alcohol consumption, unhealthy diets, insufficient physical activity and smoking, contribute to about one-third of global disability-adjusted life-years, and are leading causes of non-communicable diseases (NCDs), including cardiovascular disease,

For prevention efforts to have an impact on the general population, they need to have extensive reach among those who may benefit. No single setting will be able to achieve this, for example, only 1%–5% of individuals visiting primary healthcare clinics in Sweden are given advice with respect to their lifestyle,[5] despite many more in need of such advice. Unhealthy lifestyle behaviours also tend to cluster and interact,[6 7] for example, those who are overweight are more likely to be physically inactive, and excessive alcohol consumption may lead to weight gain. Risks from multiple unhealthy lifestyle behaviours may be multiplicative[8]; thus, it is of value to not only extend the reach of interventions, but to also investigate tools designed to support change of multiple health behaviours.

One way of reaching further into the community with a multiple health behaviour change intervention is to offer digital support tools to those searching online for help. This is especially promising in Sweden, since the internet is used daily by approximately 90% of the population, and the same proportion use smartphones on a regular basis.[9 10] A recent effectiveness trial of a digital alcohol intervention among online help-seekers in Sweden found evidence of positive effects on alcohol consumption,[11] but also that only 13.5% of study participants turned off the support, which indicates that receiving support for behaviour change through digital means is an acceptable method for many. Studies evaluating digital interventions addressing multiple health behaviours have also shown promising results.[12–15] However, the evidence of these types of interventions in more general populations is lacking, as the majority of studies have been conducted among university students, employees within specific fields, or patients with specific health conditions. In addition, behaviour interventions often consist of several components or modules, yet are commonly evaluated as a whole,[16] leaving a paucity of evidence for the effects of the dismantled components. Increasing our understanding of the effects at the component level, in particular with respect to multiple behaviours, may help move the field of behaviour interventions forward.

## Objectives

This study aims to estimate the effects of the components of a digital intervention on multiple health behaviours (alcohol, physical activity, diet and smoking) among individuals seeking help online. The objectives of the study are to:
1. Estimate the effects of a digital intervention's different components on individual health behaviours:
   a. Weekly alcohol consumption and number of episodes per month of heavy drinking.
   b. Average daily fruit and vegetable consumption.
   c. Weekly moderate to vigorous physical activity (MVPA).
   d. Four-week point prevalence of smoking.
2. Estimate the degree to which the effects of the components are mediated through perceived importance, confidence and know-how.

3. Detect interactions among health behaviour change, for example, those who stop smoking may also reduce their alcohol consumption, and the degree to which this is moderated by the components of the intervention.

## METHODS

A double-blind factorial randomised trial[17] (six factors with two levels each) will be employed to address the objectives of the study. A Bayesian group sequential design will be employed to periodically make decisions to continue or stop recruitment.[18–20] This protocol contains relevant items from the Standard Protocol Items: Recommendations for Interventional Trials.[21] The methods of this trial, including the statistical analysis plan, was preregistered on the Open Science Platform prior to enrolment commenced (https://osf.io/xyj3p/).

### Study setting, recruitment and eligibility

We will recruit individuals seeking information about health and behaviour change by advertising on Google, Bing and Facebook (restricted to Sweden), as well as on websites which focus on lifestyle and behaviour change (eg, livsstilsanalys.se). Individuals exposed to the advert will be advised to sign up to the study by sending a text message with a specific code to a dedicated phone number.

Within 10 min, individuals will receive a text message with a hyperlink that takes them to a web page with informed consent materials. Consent will be given by clicking on a button on the bottom of the page. All individuals giving informed consent will be asked to complete a baseline questionnaire, which will also assess eligibility for the trial (please see online supplemental appendix A). Individuals will be included in the trial if they fulfil at least one of five conditions:
► Weekly alcohol consumption: Consumed 10/15 (female/male) or more standard drinks of alcohol the past week. A standard drink of alcohol is in Sweden defined as 12 grams of pure alcohol.
► Heavy episodic drinking: Consumed 4/5 (female/male) or more standard drinks of alcohol on a single occasion at least once the past month.
► Fruit and vegetables: Consumed less than 500 g of fruit and vegetables on average per day the past week.
► MVPA: Spent less than 150 min on MVPA the past week.
► Smoking: Having smoked at least one cigarette the past week.

Individuals will be explicitly excluded if they do not fulfil any of the criteria or if they are less than 18 years of age. The trial information and intervention will be entirely in Swedish and delivered to participants' mobile phones, thus not comprehending Swedish well enough to sign up or not having access to a mobile phone will implicitly exclude individuals.

Åsberg K, et al. BMJ Open 2022;12:e061024. doi:10.1136/bmjopen-2022-061024

**Table 1**  Brief description of the six components of the coach intervention

| Screening and feedback | Present/absent |
|---|---|
| Every Sunday afternoon, participants will receive a text message with a hyperlink which takes them to a questionnaire regarding their current health behaviours. Once complete, feedback on their current behaviour is given in relation to national guidelines. Thereafter users are given access to the rest of the components (depending on allocation). | When absent participants will not be shown the questionnaire but instead only national guidelines without personal feedback. |
| **Goalsetting and planning** | |
| This component let participants set a goal for their future behaviour and plan for what to do when they struggle and succeed. Participants can also accept challenges for the coming week, for example, to walk for 15 min each day, or to not drink any alcohol this week. Self-composed challenges are also available. Reminders are sent via texts to participants about their goals and challenges throughout the week. | When absent, this component will not be visible. |
| **Motivation** | |
| This component contains information and tools to increase participants' motivation for change. This includes information on negative health consequences, costs induced from certain behaviours and reflective tasks. If participants choose, they can also activate motivational text messages which are sent to them throughout the week. | When absent, this component will not be visible, and text messages will not be available. |
| **Skills and know-how** | |
| Concrete tips on how to initiate and maintain change in everyday life is offered in this component. This includes giving participants strategies they can use to say no to alcoholic beverages at parties, how to increase the nutritional value of their breakfast, etc. If participants choose, they can also activate text messages with tips sent to them throughout the week. | When absent, this component will not be visible, and text messages will not be available. |
| **Mindfulness** | |
| This component aims to increase users' awareness of their own lived experience and strengthen their capacity for non-reactive, compassionate and less stressful way of being in the world. Mindfulness exercises are offered to participants, including guided meditations. | When absent, this component will not be visible, and guided meditations not available. |
| **Self-composed text messages** | |
| Participants are given the opportunity to compose messages and have them sent to themselves throughout the week (on days and times of their own choosing). A participant may for instance write a message to themselves reminding them to eat two fruits each day, to not drink anything on Wednesdays, or to go for a walk with a friend. | When absent, this component will not be visible. |

## Interventions

The digital intervention, which is called Coach, consists of six components which users access using their mobile phone, based on an intervention design we have used previously.[22 23] The intervention is designed around social cognitive theories of behaviour change, with a focus on modifying environment, intention and skills.[24 25] The intervention's components are intended to be used as a toolbox, allowing users to choose which parts of the intervention to interact with and tailor the support to their needs. Participants eligible for the trial will be allocated to one of 64 factorial conditions, each condition representing a unique combination of the six components—which are either present or absent ($2^6$=64 conditions). The intervention materials can be accessed at participants' discretion over a 4-month period, and each Sunday afternoon participants will receive a text message with a link and a reminder to access the intervention materials. A summary of the components is presented in table 1, and a detailed description of the six components is available in online supplemental appendix B.

## Outcomes
### Measures

Outcomes are listed here and subsequently explained. All questionnaires (baseline, 1-month, 2-month and 4-month follow-up) used in the trial can be found in online supplemental appendix A.

### *Primary outcome measures*

► Alcohol: Weekly alcohol consumption; monthly frequency of heavy episodic drinking.
► Diet: Average daily consumption of fruit and vegetables.
► Physical activity: Weekly MVPA.

- Smoking: Four-week point prevalence of smoking abstinence.

### Secondary outcome measures
- Perceived stress.
- Weekly consumption of sugary drinks.
- Weekly consumption of candy and snacks.
- Body mass index (BMI)
- Body mass index (BMI).
- Weekly number of cigarettes smoked.
- Quality of life (QoL).

### Mediation measures
- Importance of change.
- Confidence in one's ability to change.
- Knowledge of how to change.

### Primary and secondary outcomes
Weekly alcohol consumption will be assessed by asking participants the number of standard drinks of alcohol they consumed last week (short-term recall method[26]). Frequency of heavy episodic drinking will be assessed by asking participants how many times they have consumed 4/5 (female/male) or more standard drinks of alcohol on one occasion the past month. These two outcomes are both part of the proposed core outcome set for brief alcohol interventions,[27–29] and represent different risk behaviours which are sometimes found in the same individual and sometimes not. For instance, one may have a high weekly alcohol consumption, and thereby be at risk for negative health consequences, without consuming 4/5 or more drinks on the same occasion. Similarly, having one episode of heavy episodic drinking increases the risk of short-term consequences (such as injury) and long-term health consequences, but does not fulfil the criteria for total weekly consumption.

Diet and physical activity will be measured using a questionnaire based on the previously published questionnaire by the National Board of Health and Welfare in Sweden,[7] and was further modified to also include portion sizes. The consumption of fruit and vegetables will be measured using two questions concerning the number of portions (100 g) of fruit and vegetables (respectively) the participants ate on average per day during the past week. Sugary drinks consumption will be measured by a question regarding the number of units (33 cl) of sugary drinks participants consumed the past week, and candy and snacks will be measured using a single question regarding number of servings consumed last week. MVPA will be estimated by summing responses to two questions regarding the number of minutes spent on moderate and vigorous physical activity, respectively, during the past week.

BMI will be measured by asking participants to report their weight and height.

Four-week point prevalence of smoking abstinence (no cigarettes the past 4 weeks) will be asked as a binary question. This is a suggested measure by the Society of

| | STUDY PERIOD | | | | | |
|---|---|---|---|---|---|---|
| | Enrolment | Allocation | Post-allocation | | | Close-out |
| **TIMEPOINT** | 0 | 0 | 0 | 1 month | 2 months | 4 months |
| **ENROLMENT:** | | | | | | |
| **Informed consent** | X | | | | | |
| **Eligibility screen** | X | | | | | |
| **Allocation** | | X | | | | |
| **INTERVENTIONS:** | | | | | | |
| *Digital intervention (factorial design)* | | X | ← | | | → |
| **ASSESSMENTS:** | | | | | | |
| *Baseline questionnaire* | X | | | | | |
| *Mediator questionnaire* | X | | | X | X | X |
| *Lifestyle outcomes questionnaire* | | | | | X | X |
| *Perceived stress* | X | | | | X | X |
| *QoL* | | | | | | X |
| *Participant experience* | | | | | | X |

**Figure 1** SPIRIT figure showing participant timeline throughout the study. SPIRIT, Standard Protocol Items: Recommendations for Interventional Trials.

Research on Nicotine and Tobacco.[30] Participants who have smoked any cigarette the past 4 weeks will be asked for the number of cigarettes smoked the past week.

QoL will be measured using PROMIS Global 10,[31] both to estimate the degree to which intervention components effect QoL but also for health economic evaluations. Perceived stress will be assessed using the short form Perceived Stress Scale-4.[32]

### Mediation measures
Participants will be asked to report on confidence, importance and know-how, which are three psychosocial factors believed to be important markers of behaviour change.[24 25 33–35] To reduce participant burden, we will use single face-valid items, acknowledging the limitation of such measures.

### Participant timeline and follow-ups
A trial participant timeline is presented in figure 1. Intervention components (depending on allocation) will be made available to participants all at once and stay available to participants at their own discretion throughout the 4-month period (with weekly reminders). There are three follow-up stages: 1-month, 2-month and 4-month postrandomisation. All follow-ups will be initiated by sending text messages to participants with hyperlinks to questionnaires. The following additional attempts will be made to collect data:
1. A total of two text reminders will be sent 2 days apart to those who have not responded.
2. If there is no response to the mediator questions at the 1-month follow-up, then the questions will be sent in a text message and participants are asked to respond directly with a text.
3. If there is no response to the 2-month and 4-month follow-ups, then we will call participants to collect

Åsberg K, *et al. BMJ Open* 2022;**12**:e061024. doi:10.1136/bmjopen-2022-061024

responses for the primary outcome measures only. A maximum of five call attempts will be made.

## Assignment of interventions

Randomisation will be fully automated and computerised. Block randomisation will be used to allocate participants to the 64 conditions (random block sizes of 64 and 128). Neither research personnel nor participants will be able to influence allocation.

Research personnel will be blind to allocation throughout the trial. All participants will have access to the intervention, although with different components, and they will not be made aware of the other available conditions and will therefore be blind to allocation.

## Patient and participant involvement statement

Outcome measures used in the trial are informed by national guidelines in Sweden, as well as those set by the WHO. Also, the Swedish National Board of Health and Welfare[7] have reported that research regarding multiple health behaviour change interventions is lacking. No patients or participants were involved in the planning of this trial or design of the intervention; however, both have been informed by our previous research involving individuals looking for help to change health related behaviours.

## ANALYSIS

All analyses will be done keeping all participants in the groups to which they were randomised. Analyses will be done using both available data and imputation. Imputation will be done using multiple imputation with chained equations.[36] The implicit missing at random (MAR) assumption underlying this approach will be investigated by two attrition analyses: (1) if data are missing systematically then it may be the case that early responders (answering without reminders) differ from non-responders (requiring several attempts), and in extension that late responders are more alike non-responders. Therefore, one attrition analysis will regress primary outcomes against number of attempts to collect follow-up before a response was recorded; (2) we will further explore the MAR assumption by investigating if responders and non-responders are different with respect to baseline characteristics.

Groups will be contrasted using multilevel regression models with covariates for group by component interactions and participant level adaptive intercepts. Models of longitudinal data (primary outcomes and perceived stress) will include group by time by component interactions. We will explore pairwise interactions among components. Bayesian inference will be used to estimate the parameters of the models[37–39] (see Sample Size for priors). For each coefficient of interest, we will report the marginal posterior probability of effect, and the median will be used as a point estimate of the magnitude of the effect. We will also report on 50% and 95% compatibility intervals.

## Models

### Primary and secondary outcomes

Analyses of primary outcomes will be conducted among those fulfilling the respective criteria for inclusion at baseline, for example, weekly alcohol consumption will be analysed among those who reported having consumed 10/15 (female/male) or more units of alcohol the past week. BMI, sugary drinks, candy/snacks, QoL and perceived stress will be analysed among all participants, and number of cigarettes smoked weekly among baseline smokers.

Weekly alcohol consumption, frequency of heavy episodic drinking per month, weekly intake of candy and snacks, number of sugary drinks per week and cigarettes smoked per week are all count variables that are likely skewed and over dispersed. Therefore, these outcomes will be analysed using negative binomial regression. If found not to be over dispersed, we will consider using normal regression (possibly log transformed). Average intake of fruit and vegetables per day, MVPA minutes per week, BMI, QoL and perceived stress will be analysed using normal regression (possibly log transformed). Point prevalence of smoking abstinence will be analysed using logistic regression.

All models will be adjusted for age, sex and mediators (importance, confidence and know-how) at baseline. Primary outcomes and perceived stress will be adjusted for their respective baseline values, except for smoking prevalence which will be adjusted by the weekly number of cigarettes smoked at baseline. BMI, sugary drinks and candy/snacks will be adjusted for baseline MVPA minutes per week and average intake of fruit and vegetables per day. Number of cigarettes smoked last week will be adjusted by its baseline value. QoL will be adjusted for perceived stress at baseline.

In addition to pairwise interactions between components, effect modification will be explored in all models to assess if any of the baseline characteristics moderate the effects of the components of the intervention.

### Mediator outcomes

Mediators will be explored using a causal inference framework,[40–42] using Bayesian inference to estimate the natural direct effect and natural indirect effect (as per the definitions of Pearl[42]). We will report on the posterior distributions of these two estimates, as well as the proportion of the total effect which is accounted for by the natural indirect effect. Four models will be created for each primary outcome measure, three which investigate the mediating factors on their own, and a fourth which incorporates all mediators at once. If any baseline characteristics were found to moderate the effects in the primary analysis, then additional mediator models will be created to include these as moderators.

### Interactions among health behaviours

Outcome interactions, and determinants of such, will be investigated in an exploratory analysis. For instance, those who quit smoking may also be more likely to reduce their alcohol consumption, and this interaction may be moderated by baseline characteristics. In addition, we will investigate interactions between changes in perceived stress, QoL and behaviour change. Models to detect such interactions will be explored and findings will be used to create hypotheses for future research.

### Sample size

The trial will use a Bayesian group sequential design[18–20] to monitor recruitment with interim analyses planned for every 50 participants completing the 4-month follow-up. Each of the primary outcomes will be modelled according to the analysis plan (see the Analysis section), and coefficients for dummy variables representing presence/absence of each component at each follow-up interval will be assessed for effect, harm and futility with respect to each outcome. We let $\beta_{k,l,i}$ represent the regression coefficient for component $k$, at time $I$, for outcome $i$ and D all the data currently accumulated, then the target criteria will be:

- Effect (fruit/veg. and physical activity): $p(\beta_{k,l,i} > 0 \mid D) > 97.5\%$ and $p(\beta_{k,l,i} > 0.10 \mid D) > 50\%$.
- Harm (fruit/veg. and physical activity): $p(\beta_{k,l,i} < 0 \mid D) > 97.5\%$ and $p(\beta_{k,l,i} < -0.10 \mid D) > 50\%$.
- Effect (alcohol and smoking): $p(\beta_{k,l,i} < 0 \mid D) > 97.5\%$ and $p(\beta_{k,l,i} < -0.10 \mid D) > 50\%$.
- Harm (alcohol and smoking): $p(\beta_{k,l,i} > 0 \mid D) > 97.5\%$ and $p(\beta_{k,l,i} > 0.10 \mid D) > 50\%$.
- Futility (all outcomes): $p(-0.10 < \beta_{k,l,i} < 0.10 \mid D) > 95\%$.

Outcomes analysed using normal regression will be standardised when checking the above criteria. For the effect and harm criteria, we will use a standard normal prior for dummy covariates (mean=0, SD=1.0), and a slightly wider prior will be used for the futility criterion (mean=0, SD=2.0). The criteria should be viewed as targets, thus at each interim analysis we will evaluate each criterion and decide if we believe that recruitment should stop or continue. We will continue recruitment until one criterion is fulfilled for each component, for each outcome, at each follow-up interval. We will consider removing factors from the trial if the harm criteria are fulfilled for a component on all outcomes. We will not remove factors for which the effect or futility criteria are satisfied, as collecting additional data will facilitate reducing uncertainty regarding interaction effects. Note that we are estimating each component's effect on each outcome, thus we are not a priori excluding any combination. If a component is ineffective with respect to a specific outcome, then this will be captured by the futility criteria, and will also be reported as a finding.

While the final sample size is not determined a priori, we conducted a series of simulations with effect sizes at the minimal value of the above criteria (0.1 Cohen's d for fruit/veg and physical activity, 1.1 incidence rate ratios for alcohol and 1.1 ORs for smoking). Simulations suggested that approximately 1500–2500 participants will be necessary to recruit. However, the criteria will decide, not the simulations. Despite having more conditions than in a traditional two-arm trial (in this case 64 conditions), the factorial design is fully powered for each contrast.[17] This can be understood by observing that half the study population are given access to each individual component (see online supplemental appendix table 1 in appendix B), thus the other half creates a contrast (a type of control).

Note that the Bayesian approach allows us to make unlimited looks at the data without worrying about multiplicities and error rates, as would be necessary using a frequentist approach.[43] Also, since no fixed effect size is prespecified, we reduce the risk of stopping recruitment both too early and too late.[20]

## DISCUSSION

Maintaining a healthy diet and adequate physical exercise are proven ways to decrease the risk of many NCDs such as cancer and type II diabetes. More specifically, evidence suggests that the risk of many types of cancer is reduced by a diet which, among other things, includes vegetables and fruits and limits high-calorie foods and sugary drinks.[44] Smoking has been identified as the most prominent risk factor for developing many types of cancer, however, there are indications that more complex connections are in effect. For instance, alcohol consumption is a strong risk factor for cancer in and of itself, however, it has a synergetic relation with smoking in the context of developing certain types of cancer, meaning that a combination of these health behaviours amounts to bigger risks than their individual effects.[45 46] Research has provided strong evidence that risk factors for disease such as smoking, alcohol, physical inactivity and poor diet tend to have a clustered and co-occurring pattern in populations.[47 48] Swedish data show a similar tendency, increasing the risk of poor health outcomes in the population and hence providing additional incitement for future studies to use a multibehaviour approach. Furthermore, previous research concludes the need for future research to use a holistic approach, focusing on multiple and simultaneous interventions for behavioural change[13 47 49–52]

Two meta-analyses reported modest effects of multiple health behaviour interventions in non-clinical[50] and clinical populations,[53] with various suggested reasons, including poor implementation. Some of the limitations of past efforts may be difficult to overcome with traditional face-to-face interventions, due to the large demand on staff and other resources. Only 4 of the 69 trials in one of the meta-analyses[50] investigated the use of interventions delivered via digital technology (eg, email, text messages or websites). These trials were however limited by low power or engagement, targeted university students or young individuals, and had questionable external validity. All in all, despite the extended reach which digital interventions may have, there is a lack of evidence for digital

multiple health behaviour interventions targeting a more general population.

This factorial trial investigates the components of a novel multiple behaviour intervention. While our aim of the trial is to estimate the effects of the components on behaviour, we plan to conduct exploratory studies of engagement,[54] which in combination with effect estimates will be used to determine future directions of study. Decisions to retain or remove components will therefore not be based solely on the statistical analyses in this study, but rather combined with engagement data and the evidence from the literature more widely. If for instance some components are found to exert only small effects, but was hardly used, we are more inclined to in future studies understand why it was not used and based on this redesign the component. On the other hand, components which are used often but still exert small effects may be candidates for replacement. If some components are found to only be effective for some behaviours, then these may be candidates for inclusion among those only with these unhealthy behaviours.

## Generalisability and limitations

We have adopted a pragmatic recruitment strategy for this trial, using online channels, which closely mimics the way the intervention would be disseminated in a real-world context. The trial should therefore be viewed as estimating effectiveness of the intervention's components, rather than an efficacy. However, careful consideration should be taken due to the trial context creating expectations of and from participants,[55 56] and those who take part in trials may be systematically different from those who do not. In addition, several limitations of the trial should be considered when interpreting findings.

The factorial design of this trial allows all participants to receive some support, even if some will receive a minimal number of components. Since conditions are unknown to participants we consider them blinded to allocation, which reduces the risk of bias.[57 58] This does not however protect entirely against social desirability bias, as those who are positive to the treatment received may want to support its dissemination by reporting more positive outcomes than actual,[59] which may be less likely if fewer components of the intervention are received. Compensatory rivalry bias could exacerbate this issue.[60] We will ask questions with respect to participants' perceptions about the support received to support reasoning about the strength of these threats to validity.

Condition allocation may be revealed to research personnel when participants are called to collect follow-up data. This may be a source of bias, as non-blinded assessment of subjective measures have been found to bias estimates.[61] Deducing the exact allocation is however unlikely, and personnel are instructed to not ask about anything else than the follow-up data. Using phone calls is a strategy employed to reduce the risk of attrition bias, which we believe outweighs the risk of detection bias.

Finally, there are two methodological compromises which are important to address. First, we use single face-valid items for mediators to reduce participant burden, which means that any marked mediation effect should be carefully interpreted to relate to the full concept of importance, confidence and know-how. Second, criteria for stopping enrolment are based on the analysis of individual components which does not consider interactions among components. While it would be advantageous to include criteria for interactions, it is not practical to do so as it would increase the expected sample size markedly.

## ETHICS AND DISSEMINATION

The study was approved by the Swedish Ethical Review Authority on 2021-08-11 (Dnr 2021-02855). Participants are likely to have been motivated to sign up for the trial by the potential of receiving novel support, leading to a risk of opportunity cost if the intervention only exerts small effects on behaviour. However, considering that current prevention efforts seem to not be enough to reduce the prevalence of unhealthy behaviours, and the potential effects and reach a digital multiple health behaviour change intervention could have among those seeking help online, this risk was deemed acceptable.

Recruitment began in October 2021, and we anticipate that recruitment will last no more than 12 months. A final dataset will therefore be available in January 2023, and findings will be subsequently submitted for peer-review in open access journals.

**Contributors** Study objectives and outcomes were decided by MB, ML, PB, PH and HH. MB and KÅ designed the trial and analysis plan. Intervention materials were conceptualised and developed by KÅ, JB, MB, OL, ML, PB, PH and HH, based on an intervention design by MB. MB, KÅ and JB drafted the protocol, which was revised by ML, PB, PH, HH and OL—all authors contributed with intellectual content and approved the final version. JB, KÅ and MB will be responsible for data collection and statistical analysis. All authors will be responsible for communication of findings from the trial.

**Funding** This trial has been funded by The Swedish Cancer Society (Cancerfonden, 20 0883 Pj, PI: MB), and is an extension of the MoBILE research program which is funded by the Swedish Research Council for Health, Working Life and Welfare (Grant number 2018-01410; PI: ML).

**Competing interests** MB and PB own a private company (Alexit AB) that develops and distributes lifestyle behaviour interventions for use in healthcare settings. Alexit AB had no part in funding or planning of this trial but is relied upon for a service to send text messages.

**Patient and public involvement** Patients and/or the public were not involved in the design, or conduct, or reporting, or dissemination plans of this research.

**Patient consent for publication** Not applicable.

**Provenance and peer review** Not commissioned; externally peer reviewed.

**ORCID iD**
Marcus Bendtsen http://orcid.org/0000-0002-8678-1164

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
