## [Reviewer comments · BMJ Open]

ARTICLE DETAILS

TITLE (PROVISIONAL)	Digital multiple health behaviour change intervention targeting online help seekers: protocol for the Coach randomised factorial trial
AUTHORS	Åsberg, Katarina; Blomqvist, Jenny; Lundgren, Oskar; Henriksson, Hanna; Henriksson, Pontus; Bendtsen, Preben; Löf, Marie; Bendtsen, Marcus

VERSION 1 – REVIEW

REVIEWER	Champion, Katrina The University of Sydney, The Matilda Centre for Research in Mental Health and Substance Use
REVIEW RETURNED	14-Mar-2022

GENERAL COMMENTS	This study addresses an important public health issue and has the potential to make a meaningful contribution to the field. I have some specific comments and concerns below, primarily about the need for greater clarity and detail about the primary outcomes and intervention components. Title and abstract – consider relabelling it a ‘multiple health behaviour change intervention’ Abstract: - Please include more information about the intervention components being tested e.g how many, what are they, what are the levels. It is unclear whether the authors are dismantling an existing intervention (please mention this if so) or optimising a new intervention.- Please also include more information about the eligibility of participants e.g. age? Gender? Location?- What is the primary outcome? Or are alcohol use, diet, PA and smoking four co-primary outcomes? Introduction - As the intervention is mobile phone-based, it would be important to include some mention of mobile phone use in Sweden, effectiveness of, and engagement with, health interventions delivered via mobiles and apps, and how this study might address some of the issues for the field (low engagement, high drop out). Methods - More information about each intervention component is needed, given there is such a big emphasis on individual components in this study. Suggest adding some of the information from the appendix into the main body of text.
--

	Discussion It is unclear how the results from this study will be used to inform future research. If one component is shown to be ineffective, will it be removed from the Coach intervention? Which outcome (alcohol, diet, PA or smoking) will be used to make decisions about which components are effective/ineffective?
--	---

REVIEWER	Yahia, Najat Central Michigan University College of Education and Human Services, Human Environmental Studies
REVIEW RETURNED	16-Mar-2022

GENERAL COMMENTS	March 16, 2022 BMJ Open Dear Editor, Thank you for the invitation to review the manuscript number: 'BMJ open-2022-061024, titled: Digital Multiple Lifestyle Behavior Intervention Targeting Online Help Seekers: Protocol for The Coach Randomized Factorial Trial. This study looks interesting and beneficial. I wonder whether the authors have included the BMI of participants as a variable to look at as there is no mention in the proposal about participants' BMI. Is weight management part of the study? Examining the effectiveness of a Multidisciplinary digital program for treating obesity is important and deserves attention. Also, what changes have the authors included in this proposal compared to the study listed in citation 20, "Multiple lifestyle behavior mHealth intervention targeting Swedish college and university students: protocol for the Buddy randomized factorial trial. BMJ Open. 2021 Dec;11(12):e051044." Overall, the topic is an interesting and good study. Thank you.
---

REVIEWER	Spring, Bonnie Northwestern University Feinberg School of Medicine, Preventive Medicine
REVIEW RETURNED	21-Mar-2022

GENERAL COMMENTS	In this manuscript, Katarina Åsberg and colleagues present a complex and interesting protocol to optimize the design of a digital lifestyle behavior intervention for individuals with multiple behavioral risk factors who seek online help to establish a healthier lifestyle. Study entry criteria require enrollees to report at least one of five behavioral risk factors: heavy drinking, binge
--

drinking, smoking, low fruit/vegetable intake, low moderate-vigorous physical activity). The research design, a randomized factorial trial, allocates study participants to one of 65 conditions formed by the crossing of 6 intervention components, each of which has 2 levels. Intervention components are described (in the appendix) as: 1. Screening and feedback; 2. Goal-setting and planning; 3. Motivation; 4. Skills and know-how; 5. Mindfulness; and 6. Self-composed text messages. Recruitment is ongoing, and approximately 1500-2500 participants are to be enrolled, assessed at baseline, randomized, and re-assessed 2 and 4 months after random assignment. Interim analyses will be performed after each set of 50 participants completes 4-month follow-up to determine whether each component meets criteria for effect, harm, or futility. The plan is to continue recruitment until one of these criteria is fulfilled for each outcome (and presumably for each component?) at each follow-up interval. Intervention components (factors) will be considered for removal if the harm criteria are fulfilled. It is unclear whether they'll also be considered for removal if the 2 other criteria are met, if they're met for 1 but not both time points and for a few but not all behaviors. That decision logic could be clarified.

The rationale presented for the research is compelling and clearly stated. The high prevalence and co-occurrence of risk behaviors and the minimal attention given to addressing these by the health care system, create a clear need for low-burden, high-reach approaches to intervention. The proposed analyses are sophisticated and the ambitious timeline for trial completion is efficient. However, as presently written, aspects of the proposed research are difficult to understand.

Although the goal of the study is to optimize a multiple risk behavior intervention, the intervention components that are the main factors to be optimized (i.e., the independent variables) are not described in the main manuscript. Discussion of these components does not appear until the manuscript's appendix. Even there, the treatment components seem to be characterized chiefly in terms of Behavior Change Techniques (BCTs), but some components list multiple BCTs and others have none. It would be extremely helpful for the authors to describe one or two intervention components and how they can be applied to target several different behaviors. It would also be helpful for an international audience if the illustrations could appear in English. Also confusing is the fact that the intervention components are apparently chosen by the participant to be self-administered or not. How do the investigators conceptualize what the intervention components are, and how do they assess whether the components have been delivered with fidelity?

It is odd that the main framework applied globally to design and interpret factorial trials to optimize multi-component interventions is not mentioned: Linda Collins' multiphase optimization strategy (MOST). The approach is erroneously attributed to Susan Michie.

	What appears in the main manuscript in lieu of a description of intervention factors is text about various secondary outcomes, mediators, and potential interactions among primary outcomes (smoking and drinking). This material introduces some confusion, as it introduces secondary outcomes and analyses before operationalization of the primary aims have been clarified. Another confusing aspect is that the authors discuss their plan to interpret primarily main effects of the intervention components, eschewing interactions. They emphasize main effects chiefly in terms of supporting the argument that the study is powered sufficiently to test main component effects. It would be helpful for the authors to describe clearly their stance toward interactions involving the design's independent variables before progressing toward more esoteric discussion of potential interaction among dependent variables. Additional discussion of the primary outcome variable(s) would be helpful. Did the authors consider creating a composite measure of multiple risk behavior change, rather than what seems to be the plan – analyzing each behavioral outcome separately? Also, what is the rationale for weighting alcohol intake twice (for daily intake and again for bingeing) in the 5-item primary outcome measure? Given the proliferation of behavioral outcome variables, moderators, and intervention factors that are or are not being analyzed for interactions, it would also be helpful to hear a recap of the authors' rationale for why the study presents no problems involving multiple comparisons. Finally, more information is needed about the circumstances under which a component would be dropped and the consequences of doing so. Would this occur only if a component produced evidence of harm? What about futility? Or having achieved evidence of efficacy for a factor at one follow-up period for one behavior? Would it be necessary for a component to produce harm across all behaviors for the component to be removed? Would it be possible technically to remove a component for one but not all behaviors? Would removal of the component from data for some but not all analyses complicate interpretation? What is the planned decision-making policy for implementation at the end of the trial? Will all components not eliminated for harm, futility, or efficacy remain included in the intervention? What will be done if results differ across time and behaviors?
--	--

VERSION 1 – AUTHOR RESPONSE

REVIEWER 1

C1: This study addresses an important public health issue and has the potential to make a meaningful contribution to the field. I have some specific comments and concerns below, primarily about the need for greater clarity and detail about the primary outcomes and intervention components.

A1: Thank you, we hope that we have addressed your comments adequately below.

C2: (Title and abstract) – Consider relabelling it a ‘multiple health behaviour change intervention’

A2: The title has been changed, and the manuscript has been edited throughout to match this change.

C3: (Abstract) - Please include more information about the intervention components being tested e.g., how many, what are they, what are the levels. It is unclear whether the authors are dismantling an existing intervention (please mention this if so) or optimising a new intervention.

A3: The word limit necessarily means that we cannot give details of all the components in the abstract, however, we have clarified the number of components, added that the intervention studied is novel, and given examples of two components (goal setting and increasing motivation). We have clarified that the levels are present and absent, thus there are six factors with two level each. We removed a sentence from the background part of the abstract to make room for this change and cut down on some redundant words elsewhere in the abstract.

C4: (Abstract) - Please also include more information about the eligibility of participants e.g. age? Gender? Location?

A4: We have added to the abstract that participants will be include if they are at least 18 years of age and have at least one unhealthy lifestyle behaviour (of the four in focus in this study).

C5: (Abstract) - What is the primary outcome? Or are alcohol use, diet, PA and smoking four co-primary outcomes?

A5: We have added “(co-primary outcomes)” to the abstract to clarify that all health behaviours are our primary outcomes. Please see our response to Reviewer #3 comment C21 regarding estimating multiple effect estimates.

C6: (Introduction) - As the intervention is mobile phone-based, it would be important to include some mention of mobile phone use in Sweden, effectiveness of, and engagement with, health interventions delivered via mobiles and apps, and how this study might address some of the issues for the field (low engagement, high drop out).

A6: We have added recent data on internet and mobile-phone usage to the final paragraph of the introduction. We have also mentioned a recent study which investigated the effectiveness of a digital alcohol intervention among online help-seekers and emphasized that many found this to be an acceptable means of receiving support.

In terms of addressing some of the issues in the field regarding low engagement, we do not feel like we are in a place to do so in this study. While we will collect data on usage and will be able to analyse engagement patterns (please see our response to comment C8 below), this study is not designed to specifically address these issues, and we therefore are not comfortable promising to do so.

C7: (Methods) - More information about each intervention component is needed, given there is such a big emphasis on individual components in this study. Suggest adding some of the information from the appendix into the main body of text.

A7: Thank you for suggesting this change. We have added a table (Table 1), which describes each component in brief, with an explanation of what it means when a component is absent for some participants (depending on random allocation).

C8: (Discussion) - It is unclear how the results from this study will be used to inform future research. If one component is shown to be ineffective, will it be removed from the Coach intervention? Which outcome (alcohol, diet, PA or smoking) will be used to make decisions about which components are effective/ineffective?

A8: We will not make such decisions based on the statistical analyses from this trial alone, and not based on only the estimates of effects. Patterns of engagement using so called microscopic engagement data will be done, similar to Perski et al., to investigate which components are used more (or less) and by whom. If for instance some components are found to exert only small effects, but was hardly used, we may be more inclined to in future studies understand why it was not used and based on this redesign the component. On the other hand, components which are used often but still exert small effects may be candidates for replacement. If some components are found to only be effective for some behaviours, then these may be candidates for inclusion among those only with these unhealthy behaviours.

We have added a paragraph to the Discussion section (before the Generalisability and Limitations section) that describes our planned future research.

Perski et al. Identifying Content-Based Engagement Patterns in a Smoking Cessation Website and Associations With User Characteristics and Cessation Outcomes: A Sequence and Cluster Analysis. *Nicotine & Tobacco Research* 23(7); 2021.

REVIEWER 2

C9: This study looks interesting and beneficial.

A9: Thank you, we hope that we have addressed your comments adequately below.

C10: I wonder whether the authors have included the BMI of participants as a variable to look at as there is no mention in the proposal about participants' BMI. Is weight management part of the study? Examining the effectiveness of a Multidisciplinary digital program for treating obesity is important and deserves attention.

A10: Body mass index (BMI) is included as a secondary outcome. It is listed under Outcome -> Measures. We agree that effective interventions to treat obesity are important, however, the current intervention is focused on common health behaviours in a broad population, including non-obese individuals. Therefore, specific weight management components are not included. The intervention does emphasize increasing physical activity, reducing intake of items with high sugar content (such as sugary drinks, candy, and snacks), and the importance of a healthy diet.

C11: Also, what changes have the authors included in this proposal compared to the study listed in citation 20, "Multiple lifestyle behavior mHealth intervention targeting Swedish college and university students: protocol for the Buddy randomized factorial trial. BMJ Open. 2021 Dec;11(12):e051044."

A11: The two trials employ the same study design, i.e., both are factorial trials of a multiple behaviour intervention, but they target different populations. The trial described in the current protocol includes individuals who are searching online for help to change their health behaviours. Our previous studies in this population suggest that the mean age of participants will be around 50-55. The trial cited targets university students via their student health care centres, a younger demographic which also are in a unique transitioning time in life. The two interventions share the same basic components, but the content of the interventions have been shaped to fit each respective target group. For instance, in the intervention targeting university students, we emphasize the impact healthy behaviours may have on study results. Note that we are responsible for both trials, and that the similarities between the studies were communicated to the editor at the time of submission.

REVIEWER 3

C12: The rationale presented for the research is compelling and clearly stated. The high prevalence and co-occurrence of risk behaviors and the minimal attention given to addressing these by the health care system, create a clear need for low-burden, high-reach approaches to intervention. The proposed analyses are sophisticated and the ambitious timeline for trial completion is efficient.

A12: Thank you, we hope that we have addressed your comments adequately below.

C13: The plan is to continue recruitment until one of these criteria is fulfilled for each outcome (and presumably for each component?) at each follow-up interval. Intervention components (factors) will be considered for removal if the harm criteria are fulfilled. It is unclear whether they'll also be considered for removal if the 2 other criteria are met, if they're met for 1 but not both time points and for a few but not all behaviors. That decision logic could be clarified.

A13: Yes, the plan is to continue recruitment until one criterion is fulfilled for each outcome, component, and follow-up interval. In the first paragraph of the Sample Size section we have written: “We let $\beta_{k,l,i}$ represent the regression coefficient for component k , at time l , for outcome i , ...” and then the criteria are specified for $\beta_{k,l,i}$, and we also write “We will continue recruitment until one criterion is fulfilled for each component, for each outcome, at each follow-up interval”, also noting that these are targets that we use to decide when to stop recruitment.

In the paragraph following the target criteria, we have clarified that we will consider removing a component if it is found harmful for all outcomes. We anticipate that this is unlikely, but such findings would suggest that we should no longer expose individuals to this component. We will not remove components if found to be effective or futile, but rather continue collecting data to reduce uncertainty with respect to interaction effects among components.

C14: Although the goal of the study is to optimize a multiple risk behavior intervention, the intervention components that are the main factors to be optimized (i.e., the independent variables) are not described in the main manuscript. Discussion of these components does not appear until the manuscript's appendix. Even there, the treatment components seem to be characterized chiefly in terms of Behavior Change Techniques (BCTs), but some components list multiple BCTs and others have none. It would be extremely helpful for the authors to describe one or two intervention components and how they can be applied to target several different behaviors.

A14: In response to this comment (and Reviewer #1 comment C7), we have described the intervention components in more detail in a new table in the main document (Table 1). In the appendix, we have not listed BCTs for Component 5 (mindfulness) nor Component 6 (self-composed text messages) as neither fit into the BCT framework (we note in the appendix that we use the taxonomy when appropriate). We have listed multiple BCTs where components are multifaceted, e.g., Goalsetting and Planning which can be described by several techniques in the taxonomy.

C15: It would also be helpful for an international audience if the illustrations could appear in English.

A15: The digital support tool is in Swedish only, therefore we have no screenshots of the tool in English. We may be able to produce these, however, would like to hear from the editor regarding journal policy before doing so.

C16: Also confusing is the fact that the intervention components are apparently chosen by the participant to be self-administered or not. How do the investigators conceptualize what the intervention components are, and how do they assess whether the components have been delivered with fidelity?

A16: This is an effectiveness trials that aims to assess the effects of having access to different components of a digital intervention. Since the intervention is intended to be disseminated in a large population, comprising of thousands of individuals, part of the design of the intervention includes self-administration. This is often the case of large-scale effectiveness trials, leaning toward the pragmatic and naturalistic study of interventions. Different individuals will engage with the components in different ways, some spending more time than others on specific parts, however, we make no judgement of what the correct way is to use the components. In response to Reviewer #1 comment C8, we have added a paragraph before the Generalisability and Limitations section on how we intend

to use our findings, including the use of engagement data which gives us insights into how the different components have been used.

C17: It is odd that the main framework applied globally to design and interpret factorial trials to optimize multi-component interventions is not mentioned: Linda Collins' multiphase optimization strategy (MOST). The approach is erroneously attributed to Susan Michie.

A17: We are not adopting the MOST framework in this project, rather it is a standalone factorial trial investigating the effects of a set of components on behavioural outcomes. In MOST, one option is to use factorial trials to prune and refine components, sometimes multiple times, for later evaluation in a full-scale trial. Factorial trials in and of themselves have been used for a long time and are not novel (nor restricted) to MOST. Had we adopted MOST we would have planned for additional studies (possibly factorial), where components were refined and then evaluated in an RCT. We avoid introducing MOST in our protocol to not confuse our objectives with this study.

Since we have not mentioned MOST in our protocol, we cannot address the comment regarding the framework being erroneously attributed to Susan Michie.

C18: What appears in the main manuscript in lieu of a description of intervention factors is text about various secondary outcomes, mediators, and potential interactions among primary outcomes (smoking and drinking). This material introduces some confusion, as it introduces secondary outcomes and analyses before operationalization of the primary aims have been clarified.

A18: The protocol's sections follow the order recommended by SPIRIT, in which outcomes (and measures) are presented after interventions but prior to participant timeline and statistical analyses. We deviated from SPIRIT by moving the Sample Size section to after the Analysis section, as it is hard to follow without first reading about the planned analyses. Perhaps the added description of the intervention content in Table 1 satisfies this comment by better explaining how the intervention is intended to work before moving on to the outcomes.

C19: Another confusing aspect is that the authors discuss their plan to interpret primarily main effects of the intervention components, eschewing interactions. They emphasize main effects chiefly in terms of supporting the argument that the study is powered sufficiently to test main component effects. It would be helpful for the authors to describe clearly their stance toward interactions involving the design's independent variables before progressing toward more esoteric discussion of potential interaction among dependent variables.

A19: We do not plan to eschew interactions between components. In the Analysis section, second paragraph, we write that "We will explore pairwise interactions among components". In the Discussion section, last paragraph, we mention that we will not consider interactions among components with respect to when to stop trial recruitment. We will however estimate interaction effects among components. We have added a sentence to the final paragraph of the Analysis -> Models -> Primary and Secondary Outcomes section, to again emphasize that we will estimate pairwise interactions between components. Also, since this trial does not adopt a null hypothesis framework, we do not run any tests, and therefore are not concerned about power in the traditional sense of the word (see for instance Berry or Bendtsen).

Berry D. Bayesian clinical trials. *Nat Rev Drug Discov* 5, 2006;27–36.

Bendtsen M The P Value Line Dance: When Does the Music Stop? *J Med Internet Res* 2020;22(8):e21345

Bendtsen M. A Gentle Introduction to the Comparison Between Null Hypothesis Testing and Bayesian Analysis: Reanalysis of Two Randomized Controlled Trials. *J Med Internet Res* 2018;20(10):e10873.

C20: Did the authors consider creating a composite measure of multiple risk behavior change, rather than what seems to be the plan – analyzing each behavioral outcome separately? Also, what is the rationale for weighting alcohol intake twice (for daily intake and again for bingeing) in the 5-item primary outcome measure?

A20: We decided against a composite score as it does not allow for investigation of the effects of different components on different behaviours. Differences in specific behaviours between groups at follow-up can be interpreted considering the evidence we have about, for instance, the effects on future health from reducing one’s drinking. A difference in a composite score between groups is difficult to interpret, as it has no real-world meaning, and can be induced by differences in different behaviours (i.e., one individual may have improved their composite score by reducing their alcohol consumption, while another may have done so by quitting smoking).

The rationale for two alcohol measures is that they represent two different behaviours (sometimes found in the same individual, and sometimes not). For instance, drinking four drinks per month is not considered a major health risk, unless they are all consumed during one occasion. Similarly, one can have a high weekly consumption without ever passing four drinks on one occasion, and therefore be subject to risk from alcohol consumption despite never being overly intoxicated. This is also the rationale for why both are part of the core outcome set for brief alcohol interventions. We have added a few sentences towards this to the first paragraph of the Primary and Secondary Outcomes section.

C21: Given the proliferation of behavioral outcome variables, moderators, and intervention factors that are or are not being analyzed for interactions, it would also be helpful to hear a recap of the authors’ rationale for why the study presents no problems involving multiple comparisons.

A21: As the protocol is already quite lengthy, we will give some comments on this here, but not make any changes to the manuscript. We have mentioned in the final paragraph of the Sample Size section that error rates are not a concern in the statistical approach that we are taking, with reference to the literature. The literature on Bayesian vs. frequentists statistics is deep and has a long history. A recommended start is here: Bendtsen M. A Gentle Introduction to the Comparison Between Null Hypothesis Testing and Bayesian Analysis: Reanalysis of Two Randomized Controlled Trials. *J Med Internet Res* 2018;20(10):e10873.

In the frequentists approach, where null hypothesis testing is highly prevalent, one is interested in a “true” and fixed value of the effect estimate. This can be thought of as the most likely effect estimate that you would get if you were to redo your trial an infinite number of times. A null hypothesis test assumes that this “true” and fixed effect is exactly the null, i.e., that the intervention has exactly no effect, and then we calculate the probability that we observed what we did (i.e., our effect estimate). If the probability is “low” (conventionally, 0.05), then we reject our hypothesis that the “true” and fixed effect is the null – concluding a statistically significant finding. Of course, there is a chance that you are wrong despite a “low” probability, and this is a concern when one sees “multiple testing”, i.e., you are increasing the number of times you may be wrong, compounding the probability. If one keeps

buying tickets in a raffle, at some point one is going to win (or lose as it were).

We are adopting a Bayesian inference framework to estimate the posterior distribution of effect estimates, which incorporates the information available in the data that we have collected in this trial and our prior only. In other words, we are presenting our findings in the form of a distribution over effect estimates which shows their relative compatibility with the data that we have collected (given the prior). We could for instance stop after already 5 participants and present the compatibility of different effects with our data. The uncertainty surrounding our effect estimates would be high, but nonetheless informative given the data available. The prior distribution that we give our effect estimates also ensures that any presentation of posterior distributions is not dominated by only these 5 participants, but also about our prior beliefs (which we have encoded as sceptical with standard normal priors). We start with a belief that there are only very small effects, and let the data tell us otherwise. With this in mind, we are not making any claims about estimates that may or may not have been observed if the trial was conducted an infinite number of times. We do not make an assertion about a “true” and fixed value, and then with some probability reject it – rather we only present the information that is contained in the data collected. This is part of why a discussion about “multiple testing” and error rates do not apply in a Bayesian framework – in a sense, there is no probability of being wrong because we are not asserting anything.

It should be noted that the criteria we use for stopping recruitment are not assessments of “true” and fixed effects. They are targets that help us decide if the uncertainty surrounding an effect estimate, taking into account the prior and data, have been shaped enough that we are comfortable to stop the trial and present our findings.

Bayesian inference is about presenting a posterior distribution of effects which can then be used for scientific inference, which requires understanding of many different factors beyond the effect estimate itself – including other risks in disseminating interventions to the public, budgets and scalability, opportunity costs, etc. This means that any estimate of the posterior distribution of effects is informative, and not only estimates that cross an arbitrary statistical threshold.

C22: Finally, more information is needed about the circumstances under which a component would be dropped and the consequences of doing so. Would this occur only if a component produced evidence of harm? What about futility? Or having achieved evidence of efficacy for a factor at one follow-up period for one behavior? Would it be necessary for a component to produce harm across all behaviors for the component to be removed? Would it be possible technically to remove a component for one but not all behaviors? Would removal of the component from data for some but not all analyses complicate interpretation?

A22: This seems to be a duplicate (at least partially) of a previous comment (C13). We hope that the response (A13) to that comment also satisfies the points raised here. Since we are planning to remove a component only if it is found to be harmful for all outcomes, its removal will not affect nor complicate interpretation.

C23: What is the planned decision-making policy for implementation at the end of the trial? Will all components not eliminated for harm, futility, or efficacy remain included in the intervention? What will be done if results differ across time and behaviors?

A23: In response to Reviewer #1 comment C8, we have added a paragraph before the

Generalisability and Limitations section on how we intend to use our findings, including the use of engagement data which gives us insights into how the different components have been used.

VERSION 2 – REVIEW

REVIEWER	Champion, Katrina The University of Sydney, The Matilda Centre for Research in Mental Health and Substance Use
REVIEW RETURNED	01-Jun-2022
GENERAL COMMENTS	Thank you - all of my comments have now been addressed.